# Implementation of 2% Chlorhexidine Bathing to Reduce Healthcare-Associated Infections Among Patients in the Intensive Care Unit

**DOI:** 10.3390/microorganisms13010065

**Published:** 2025-01-02

**Authors:** Hsu-Liang Chang, Tzu-Ying Liu, Po-Shou Huang, Chin-Hwan Chen, Chia-Wen Yen, Hui-Zhu Chen, Shin-Huei Kuo, Tun-Chieh Chen, Shang-Yi Lin, Po-Liang Lu

**Affiliations:** 1Department of Internal Medicine, Kaohsiung Municipal Ta-Tung Hospital, Kaohsiung Medical University, Kaohsiung 801, Taiwan; hsuliac@gmail.com (H.-L.C.); 1000420kmuh@gmail.com (S.-H.K.); amoe616@gmail.com (S.-Y.L.); 2Department of Internal Medicine, Kaohsiung Medical University Hospital, Kaohsiung Medical University, Kaohsiung 807, Taiwan; idpaul@gmail.com; 3Infection Control Office, Kaohsiung Municipal Ta-Tung Hospital, Kaohsiung Medical University, Kaohsiung 801, Taiwan; ned740206@gmail.com; 4Department of Nursing, Kaohsiung Municipal Ta-Tung Hospital, Kaohsiung Medical University, Kaohsiung 801, Taiwan; 1028199@kmuh.org.tw (P.-S.H.); 1038172@kmuh.org.tw (C.-H.C.); 1048093@kmuh.org.tw (C.-W.Y.); sakura.huizhu@gmail.com (H.-Z.C.); 5Graduate Institute of Medicine, College of Medicine, Kaohsiung Medical University, Kaohsiung 807, Taiwan; 6School of Medicine, College of Medicine, Kaohsiung Medical University, Kaohsiung 807, Taiwan; 7Center for Tropical Medicine and Infectious Disease Research, Kaohsiung Medical University, Kaohsiung 807, Taiwan; 8Center for Medical Education and Humanizing Health Professional Education, Kaohsiung Medical University, Kaohsiung 807, Taiwan; 9Center for Liquid Biopsy and Cohort Research, Kaohsiung Medical University, Kaohsiung 807, Taiwan

**Keywords:** healthcare-associated infections, catheter-associated urinary tract infections, catheter-associated bloodstream infections, chlorhexidine gluconate, multidrug-resistant organisms

## Abstract

Healthcare-associated infections (HAIs) significantly increase morbidity, mortality, length of hospital stays, and costs, particularly among ICU patients. Despite standard interventions, catheter-associated urinary tract infections (CAUTI) and central line-associated bloodstream infections (CLABSI) remain major HAI contributors. This study evaluated the efficacy of daily 2% chlorhexidine gluconate (CHG) bathing in reducing HAI incidence, specifically CAUTI, CLABSI, and multidrug-resistant organisms (MDROs), in a 20-bed ICU at a regional hospital. Using a prospective, uncontrolled before-and-after design, we compared traditional soap-water bathing (pre-intervention period) with CHG bathing over a one-year intervention and one-year post-intervention follow-up. The total number of patients and patient days admitted to the ICU per year were around 1330–1412 patients and 6702–6927 patient days, respectively, during 2018–2020. Results showed a significant reduction in HAI incidence rates from 3.43‰ to 0.58‰ (*p* < 0.05) during the intervention and sustained benefits post-intervention. Incidences of CAUTI and CLABSI decreased markedly (*p* < 0.05), with reduced MDRO isolates, including methicillin-resistant *Staphylococcus aureus*, vancomycin-resistant *Enterococci*, carbapenem-resistant *Acinetobacter baumannii*, and *Pseudomonas aeruginosa*. Our findings support the implementation of daily CHG bathing as an effective strategy to reduce HAI and MDROs in ICU settings.

## 1. Introduction

Healthcare-associated infections (HAIs) can increase the mortality and morbidity of hospitalized patients, the length of stay, and medical costs. Patients in the intensive care unit (ICU) have the highest risk of HAI. According to the statistics from the Taiwan Centers for Disease Control in 2019, the HAI densities of ICU were 6.0‰ and 4.8‰ in medical centers and regional hospitals, respectively. Although care bundles were applied to reduce HAI, catheter-associated urinary tract infections (CAUTI) and central line-associated bloodstream infections (CLABSI) are still the leading causes of HAI [1]. Daily chlorhexidine digluconate (CHG) bathing and short-course nasal mupirocin have been reported, which can effectively decrease bacteriuria and candiduria in male patients in the ICU [2].

Chlorhexidine is a biguanide cationic antiseptic molecule, which has been used in a variety of applications for HAI prevention, including routine handwashing, pre-procedure skin preparation, indwelling catheter exit-site care, oral care for prevention of ventilator-associated pneumonia, and whole-body bathing [3]. CHG is available in a range of concentrations from 0.05% to 4% (*w*/*v*) in aqueous solutions and in combination with different alcohols. CHG has broad-spectrum, non-sporicidal antimicrobial activity against Gram-positive and Gram-negative bacteria, yeasts, and some lipid-enveloped viruses, including HIV. CHG is generally more active against Gram-positive bacteria than Gram-negative bacteria [4]. The positively charged CHG molecule is attracted to the negatively charged phospholipids in the bacterial cell wall. At low concentrations (<0.5%), CHG is bacteriostatic, altering the cell wall and leading to loss of cell membrane integrity and leakage of intracellular components. At higher concentrations (≥0.5%), CHG is bactericidal, causing cell death following coagulation of the cytoplasmic components and precipitation of proteins and nucleic acid [5]. Decolonization with topical CHG is superior to regular soap because it binds to skin proteins and continues to exert its antiseptic activities on the skin for up to 24 h [6].

Several meta-analyses of the effects of CHG bathing on reducing HAI have been reported, and the results showed that CHG bathing could prevent central line-associated bloodstream infections, especially Gram-positive bacteria (including methicillin-resistant *Staphylococcus aureus* and vancomycin-resistant *Enterococci*) [7]. However, the effects of catheter-associated urinary tract infections and Gram-negative bacteria are controversial [5,6]. Four systematic reviews revealed no significant change in risk reduction in overall or catheter-associated urinary tract infections [7,8,9,10]. One report favored daily CHG bathing as beneficial for preventing catheter-associated urinary tract infections [11]. Therefore, the present study aimed to investigate the efficacy of daily 2% CHG bathing for healthcare-associated infections (including central line-associated bloodstream infections and catheter-associated urinary tract infections) and multidrug-resistant microorganisms.

## 2. Materials and Methods

### 2.1. Study Design and Setting

The prospective, uncontrolled before-and-after study was performed at a 400-bed regional hospital in 2019 in southern Taiwan, which had a 20-bed mixed medical and surgical ICU. This study was approved by the Institutional Ethics Committee of Kaohsiung Medical University Hospital (approval no. KMUHIRB-E(I)-20180084). This study included all patients treated in the ICU, excluding patients with a history of hypersensitivity to CHG. The ICU physician decided whether the patient with multiple wounds or pressure sores was recruited in the study.

### 2.2. Intervention

1. Pre-intervention period (year 2018): The daily hygiene procedure was performed using traditional liquid soap-water bathing, including the perineal area.

2. Intervention period (year 2019): Owing to the anionic surfactant interfering with the disinfection efficacy of CHG, we used a specific anionic surfactant non-containing cleaning solution to replace traditional liquid soap before the CHG bath. Two nurses, as one group, performed the CHG bathing procedure with 2% CHG-impregnated wipes: Step 1: Anterior surface from neck to upper abdomen; Step 2: Right arm and palm; Step 3: Left arm and palm; Step 4: Lower abdomen, groin, and perineum; Step 5: Right leg and foot; Step 6: Left leg and foot; Step 7: Back surface from neck to anus; Step 8 (optional): External surface of urinary catheter 15 cm in length from urethral meatus. Because the CHG had potential toxicity to the eyes and central nervous system, we prevented contact with the eyes and external auditory canal with 2% CHG (Figure 1).

Before introducing the new CHG bathing procedure, we updated the standard operating procedure of daily baths for ICU patients. We made the standard procedure learning video for all the ICU nursing staff. We also checked the procedure’s correctness and monitored the procedure adherence by three senior nursing staff before and during the intervention period. This study was started on 1 January 2019 and maintained for one year.

3. Post-intervention period (year 2020): Owing to no obvious adverse events, the updated 2% CHF bathing procedure was continued after the study without regular external monitoring of the procedure adherence.

The demographic data were recorded, including Acute Physiology and Chronic Health Evaluation (APACHE II) score, urinary catheter and central venous catheter utilization, microbiological data, and adverse events associated with 2% CHG. The incidence rates (‰, events/1000 patient-days) of healthcare-associated infections were diagnosed according to the Taiwan Centers for Diseases Control guideline, which was revised from the definition of the Centers for Disease Control and Prevention, United States [12].

Catheter-associated urinary tract infection (CAUTI) is defined as a urinary tract infection occurring in a patient with an indwelling urinary catheter that had been in place for more than 2 calendar days on the date of the event, with the catheter still in place on the event date or the day before. The infection must meet the following criteria: the patient exhibits at least one of the following signs or symptoms (fever >38 °C, suprapubic tenderness, costovertebral angle pain or tenderness, dysuria, urgency, or frequency) with no other recognized cause and has a positive urine culture showing at least 10^5^ colony-forming units (CFU)/mL of no more than two organisms.
CAUTI Rate per 1000 catheter-days = (Number of CAUTI cases/Numbers of urinary catheter-days) × 1000

Catheter-associated bloodstream infection (CLABSI) is defined as laboratory-confirmed bloodstream infection (LCBI) occurring in a patient with a central catheter that had been in place for more than 2 calendar days on the date of the event, with the catheter still in place on the event date or the day before. The LCBI must meet one of the following criteria:

1. The patient has a recognized pathogen identified from one or more blood cultures, and the pathogen is not attributable to an infection at another site.

2. The patient exhibits at least one of the following signs or symptoms: fever (>38 °C), chills, or hypotension, and a common skin contaminant (e.g., coagulase-negative staphylococci) is identified from two or more blood cultures drawn on separate occasions, and the infection is not related to another site.
CLABSI Rate per 1000 central line-days = (Number of CLABSI cases/Numbers of central line-days) × 1000

### 2.3. Statistical Analysis

All data were analyzed using SPSS version 20.0 (IBM Corp, Armonk, NY, USA). Continuous variables among different causal pathogens were analyzed with the nonparametric Kruskal–Wallis test. Categorical variables were compared with the chi-square test. When more than 20% of cells have expected frequencies < 5, Fisher’s exact test was applied. A *p*-value of <0.05 was considered statistically significant (two-tailed).

### 2.4. Systematic Review and Meta-Analysis

#### 2.4.1. Database Search

Two authors (Chang HL and Kuo SH) independently conducted a literature search in the PubMed, EMBASE, and Cochrane Library databases up to 6 November 2024, using medical subject headings and free terms to capture “chlorhexidine” and “urinary tract infection”. No restrictions were applied regarding publication language or year. 

#### 2.4.2. Study Selection and Eligibility

After removing duplicated studies, the titles and abstracts of all identified studies were screened for eligibility. We included randomized controlled trials (RCTs) and before-and-after studies that compared outcomes of the prevention of UTI by chlorhexidine bathing in the ICU. Studies were excluded if the study did not include ICU patients, lacked specific infection rate data, or were only microbiology data, narrative reviews, conference abstracts, study protocols, or case reports. 

#### 2.4.3. Data Extraction

Two authors (Chang HL and Kuo SH) independently extracted data on the first author, publication year, number of patients, study design, and study outcomes. The third author (Chen TC) will make a final conclusion if there are any inconsistent data extraction results.

#### 2.4.4. Bias Assessment

Quality assessment will be conducted by two authors (Chang HL and Kuo SH). The included RCTs will be reviewed using the Cochrane Risk of Bias (ROB 2.0) Assessment Tool. The RCTs will be evaluated for their potential biases, such as allocation bias, performance bias, attrition bias, detection bias, and reporting bias. The included observational studies will be reviewed using the Newcastle–Ottawa Scale (NOS). This review will consider potential confounding factors in the research process, such as selection, comparability, and exposure to the outcomes. If there is a disagreement between the two researchers, a third author (Chen TC) will join the discussion to reach a consensus.

#### 2.4.5. Statistical Analysis of Meta-Analysis

We calculated incidence rate ratios (IRR) with 95% confidence intervals (CI) using a random-effects model, applied separately to RCTs and before-and-after studies. Statistical heterogeneity was evaluated with the I^2^ value, the Cochrane χ^2^-test (Q-test), and *p* < 0.05, considered statistically significant. The forest plot was made by using R version 4.4.1 with the ‘meta’ package. Publication biases were assessed by funnel plots and Egger’s test.

## 3. Results

### 3.1. Participants’ Characteristics

The demographic characteristics and patient severity were not significantly changed among the three study periods, especially during the COVID-19 pandemic in 2020. The utilization of urinary and central venous catheters was also not significantly different during the three study periods (Table 1). No adverse events related to CHG bathing were observed during the intervention and post-intervention periods.

### 3.2. Healthcare-Associated Infections

The healthcare-associated infection incidence rates in the ICU were significantly reduced after the intervention, and the efficacy was preserved during the post-intervention period (baseline period: 3.43‰; intervention period: 0.58‰, *p* < 0.05 compared with baseline; post-intervention period: 1.59‰, *p* < 0.05 compared with baseline). Overall urinary tract infection (baseline: 2.09‰ vs. intervention: 0.43‰, *p* < 0.05) and catheter-associated urinary tract infection incidence rates (baseline: 2.09‰ vs. intervention: 0.28‰, *p* < 0.05) were significantly reduced during the intervention period in comparison to the baseline period. The incidence rates of urinary tract infection and catheter-associated urinary tract infection were raised slightly (both: 0.87‰) during the post-intervention period without statistical significance between the intervention periods. Overall bloodstream infection (baseline: 1.19‰ vs. intervention: 0.14‰, *p* < 0.05) and catheter-associated bloodstream infection incidence rates (baseline: 1.18‰ vs. intervention: 0.00‰, *p* < 0.05) were significantly reduced during the intervention period in comparison to the baseline period. The incidence rates of bloodstream infection and catheter-associated bloodstream infection were raised slightly (both: 0.72‰) during the post-intervention period without statistical significance between the intervention periods (Figure 2 and Table 2).

### 3.3. Multidrug-Resistant Microorganisms

The bacterial isolates number of multidrug-resistant microorganisms was also significantly reduced, not only Gram-positive bacteria (methicillin-resistant *Staphylococcus aureus* and vancomycin-resistant *Enterococci*, 13 to 1) but also Gram-negative bacteria (carbapenem-resistant *Acinetobacter baumannii*, carbapenem-resistant *Pseudomonas aeruginosa*, and carbapenem-resistant *Enterobacterales*, 35 to 12) after the intervention. The bacterial isolate number of multidrug-resistant microorganisms from sputum, urine, and blood specimens was also significantly reduced after intervention (sputum: 23 to 11; urine: 4 to 0; blood: 14 to 2, respectively). The effects can be persistent till the post-intervention period (Table 3).

### 3.4. Systematic Review and Meta-Analysis

We added newly published clinical studies (including randomized control trials and before-and-after studies) and our current study results to generate new meta-analysis results [13,14,15,16,17,18,19,20,21,22,23,24,25]. All studies were appraised by the ROB 2.0 assessment tool and Newcastle–Ottawa Scale, and they were rated as moderate to high quality. The meta-analysis showed the overall incidence rate ratio (IRR) of prevention UTI was 0.82 (95% confidence interval (C.I.) 0.67–0.99) by random effect model (randomized control trials: 0.80 [95% C.I. 0.63–1.01] and before-and-after studies: 0.79 [95% C.I. 0.55–1.13], respectively), which indicated a marginal effect on reducing the incidence of urinary tract infection (Figure 3). Visual analysis of the funnel plot and Egger’s test did not show a high risk of publication bias.

## 4. Discussion

Daily 2% CHG bathing can significantly reduce the overall incidence of healthcare infections, catheter-associated urinary tract infections, and central line-associated bloodstream infections among ICU patients. The number of multidrug-resistant microorganism infections also decreased. No adverse events related to CHG bathing were observed. Although the infection incidence rates rose slightly during the post-intervention period, the efficacy was preserved after the intervention without intensive monitoring.

Several systematic reviews and meta-analysis reports have addressed the issue of implementing CHG bathing for ICU patients to prevent urinary tract infections. The inclusion criteria (only randomized control trials or combined observational studies) and analysis method of outcomes were various among these reports. We added newly published clinical studies (including randomized control trials and before-and-after studies) and our current study results to generate new meta-analysis results [13,14,15,16,17,18,19,20,21,22,23,24,25]. The meta-analysis showed the overall incidence rate ratio (IRR) of prevention UTI was 0.82 (95% confidence interval (C.I.) 0.67–0.99) by random effect model (randomized control trials: 0.80 [95% C.I. 0.63–1.01] and before-and-after studies: 0.79 [95% C.I. 0.55–1.13], respectively), which indicated a marginal effect on reducing the incidence of urinary tract infection (Figure 3). The prior meta-analysis focusing on daily CHG bathing to reduce Gram-negative infections was insignificant [26]. Why did our study show a 79% risk reduction in urinary tract infection incidence (IRR = 0.21, 95% C.I.: 0.06–0.72)? Although cleaning the periurethral area with antiseptics is not recommended to prevent catheter-associated urinary tract infections, routine hygiene is recommended according to current guidelines. The proposed mechanism of CHG for preventing catheter-associated urinary tract infections was the reduction in microbial flora in the perineum, periurethral tissue, and urinary catheter surface by the prolonged and residual effects of CHG, which reduced the risk of colonization and subsequent infection from manipulation of urinary catheters [6,27]. Generally, the 4% CHG solution was applied and then rinsed off in the shower, while 2% CHG is used as a leave-on product (CHG-impregnated clothes or wipes) for bathing. The 2% CHG leave-on product is favored because it results in high residual concentrations of CHG on the skin, which can provide germicidal activity for up to 24 h [6]. The presence of organic material reduces the effectiveness of CHG. Additionally, the anionic surfactant would interfere with the disinfectant efficacy of CHG. We used a specific anionic surfactant non-containing cleaning solution instead of traditional liquid soap to remove excess organic substance on the skin surface before CHG bathing, which may enhance the effect of CHG. We also reinforced the cleaning of the perineum with the additional cleaning on the external surface of the urinary catheter 15 cm in length from the urethral meatus, which may have had the effect of meatal cleaning [28]. The advantages of our study included the package of 2% CHG-impregnated clothes, a specific anionic surfactant non-containing cleaning solution, and enhanced perineum cleaning with additional cleaning on the external surface of the urinary catheter 15 cm in length from the urethral meatus. Further well-designed randomized control trials are needed to confirm the effectiveness of daily CHG bathing for preventing catheter-associated urinary tract infections.

Chlorhexidine gluconate daily bathing serves as a critical preventive strategy in ICUs, targeting both urinary tract infections and infections caused by Gram-negative bacteria. Mechanistically, chlorhexidine exhibits broad-spectrum antimicrobial properties by disrupting bacterial cell membranes, leading to leakage of cellular contents and bacterial death. This effect is particularly important in reducing the skin colonization of pathogens such as *Escherichia coli*, *Klebsiella pneumoniae*, *Pseudomonas aeruginosa*, and *Acinetobacter baumannii*, which are common culprits in Gram-negative bacterial infections and CAUTIs [26,29,30]. Additionally, chlorhexidine impairs biofilm formation on the skin and around catheter insertion sites, further minimizing the risk of bacterial migration and subsequent infection. In terms of efficacy, studies have demonstrated that chlorhexidine bathing significantly reduces the incidence of CAUTIs by lowering bacterial load near the urethral meatus and surrounding catheter areas [28]. It also mitigates the risk of bloodstream infections caused by Gram-negative organisms, particularly in patients with indwelling urinary catheters or central lines. By disrupting the transmission of pathogens and enhancing infection control, chlorhexidine has proven to be an effective measure for reducing healthcare-associated infections in ICU settings [31,32]. However, maintaining adherence to daily bathing protocols is essential to ensure its efficacy in clinical practice.

Compliance also plays a vital role in the effectiveness of CHG bathing. Assessing and monitoring adherence and skillful quality are crucial for success. We did the quality assessment after staff training and before clinical application, which made the implementation of daily CHG bathing for ICU patients a great success. Still, the healthcare-associated infection incidence rates slightly rose during the post-intervention period, which reflected that intensive monitoring and annual refresher training were necessary. However, we did not assess chlorhexidine susceptibility in pathogens isolated during the post-intervention period (year 2020). Therefore, the possibility of chlorhexidine resistance contributing to the increased HAI incidence cannot be entirely excluded.

The widespread use of CHG has raised concerns about resistant emergence and cross-resistance to other antimicrobials. Wand et al. showed that the adaptation of *Klebsiella pneumoniae* isolates to CHG exposure led to an increase in minimal inhibitory concentration values for colistin from 2–4 mg/L to >64 mg/L [33]. Not only did Gram-negative bacteria show increased resistance to chlorhexidine, but also the biocide-resistant genes (*qac*A/B or *smr*) positive methicillin-resistant *Staphylococcus aureus* isolates were identified from chlorhexidine-impregnated catheter-related bloodstream infections [34]. Intrinsic microbial resistance to CHG is partly due to bacterial degradative enzymes and cellular impermeability. It explained the reason for the lower CHG susceptibility of Gram-negative bacteria. Both Gram-positive and Gram-negative bacteria with reduced CHG susceptibility have been identified, partly due to efflux pumps. However, inconsistencies in testing methods (including phenotype and genotype) make it difficult to assess the broader impact of CHG use on resistance. Despite these concerns, most evidence suggests that the risk of significant bacterial resistance from CHG is low and should not be prevented from its use in preventing healthcare-associated infections. Continued research and standardized testing are needed, but the benefits of CHG in clinical settings remain great [4,35].

Our study results showed no adverse events related to CHG bathing were observed. However, adverse reactions to CHG involve immediate IgE-mediated type I hypersensitivity and delayed cutaneous T cell-mediated type IV hypersensitivity, ranging from mild contact dermatitis to anaphylaxis or death, especially due to CHG-coating central venous catheters. Pre-procedure compatibility checks, history taking, and increased awareness of CHG hypersensitivity should be cautious while implementing CHG daily bathing in clinical practice [36].

The present study had several limitations. First, there were only 20 beds in our ICU. We were unable to perform a randomized control trial, and we used a before-and-after study design with historical control. Second, this is only a single-center, small-sample study. However, this study had some strengths. We applied a specific anionic surfactant non-containing cleaning solution before 2% CHG daily bathing and enhanced perineum cleaning with additional cleaning on the external surface of the urinary catheter 15 cm in length from the urethral meatus. The intervention was started after well-trained and certified nursing staff. They all contributed to our study’s significant reduction in catheter-associated urinary tract infection incidence.

## 5. Conclusions

The present study demonstrated the implementation of daily 2% CHG bathing can significantly reduce the overall healthcare infection incidence rates, central line-associated bloodstream infections, and catheter-associated urinary tract infections among ICU patients, which was controversial in the prior meta-analyses. The number of multidrug-resistant microorganism infections also decreased. Further well-designed, multicenter randomized control trials are needed to consolidate the effectiveness of daily CHG bathing for preventing catheter-associated urinary tract infections.

## Figures and Tables

**Figure 1 microorganisms-13-00065-f001:**
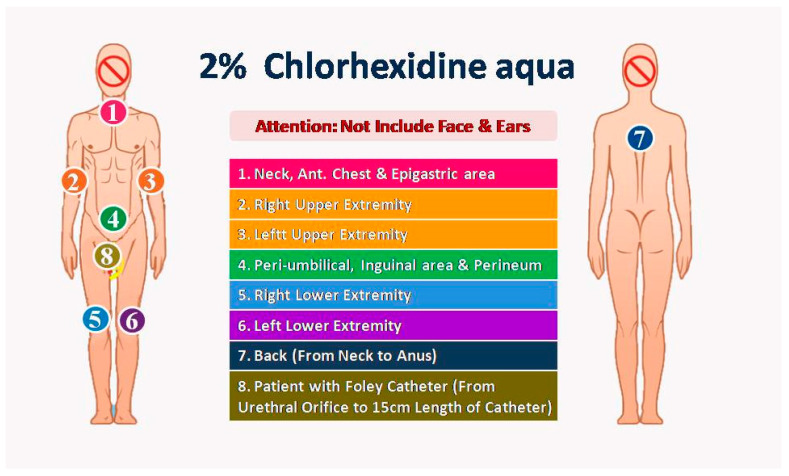
The processes of 2% chlorhexidine daily bathing.

**Figure 2 microorganisms-13-00065-f002:**
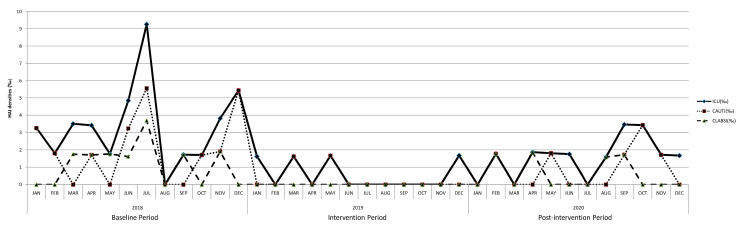
The healthcare-associated infection densities and catheter-associated infection densities over the study periods.

**Figure 3 microorganisms-13-00065-f003:**
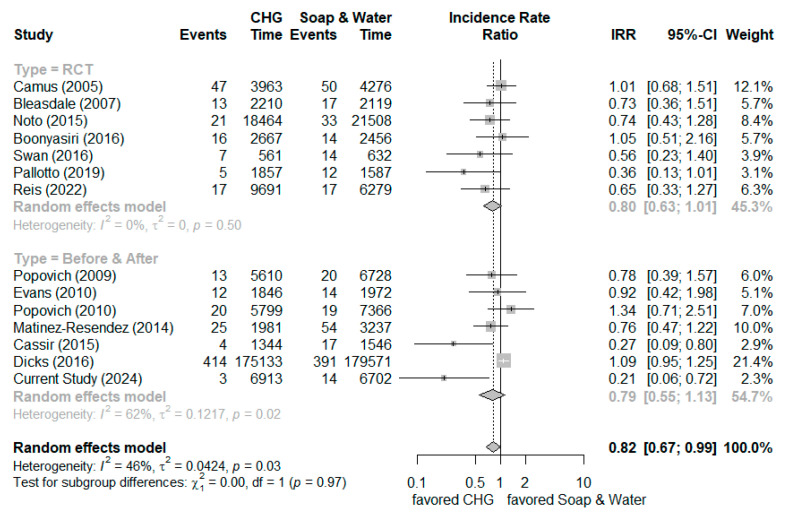
The forest plot of meta-analysis on the effectiveness of chlorhexidine bathing for preventing urinary tract infections [13,14,15,16,17,18,19,20,21,22,23,24,25].

**Table 1 microorganisms-13-00065-t001:** The characteristics and parameters over the study periods.

Characteristics	Baseline Period (Year 2018)	Intervention Period (Year 2019)	Post-Intervention Period (Year 2020)
Numbers of patients	1330	1412	1372
Total patient-days	6702	6913	6927
Gender (Male%)	58.2	58.8	60.15
Age	69.75 ± 14.99	68.84 ± 24.74	69.80 ± 15.27
Severity (APACHE II score)	14.16 ± 8.09	12.84 ± 7.40	13.19 ± 7.34
Urinary catheterization rates (%)	63.62	63.31	70.58
Urinary catheterization duration (d)	4.64 ± 0.77	4.43 ± 0.67	4.39 ± 0.53
Central-line catheterization rates (%)	40.95	42.71	45.58
Length of stay (d)	5.13 ± 0.42	4.94 ± 0.54	5.06 ± 0.42
Mortality rates (%)	13.5	13.4	13.6

**Table 2 microorganisms-13-00065-t002:** The incidence rates of healthcare-associated infections.

Characteristics	Baseline Period (Year 2018)	Intervention Period (Year 2019)	Post-Intervention Period (Year 2020)
Healthcare-associated infection incidence rates (‰) and (numbers)	3.43 (23)	0.58 * (4)	1.59 ^#^ (11)
Urinary tract infection incidence rates (‰) and (numbers)	2.09 (14)	0.43 * (3)	0.87 (6)
Bloodstream infection incidence rates (‰) and (numbers)	1.19 (8)	0.14 * (1)	0.72 (5)
Catheter-associated urinary tract infection incidence rates (‰) and (numbers)	2.09 (14)	0.28 * (2)	0.87 (6)
Central-line-associated bloodstream infection incidence rates (‰) and (numbers)	1.19 (8)	0.00 * (0)	0.72 (5)

* *p* < 0.05, comparison between intervention period vs. baseline period. ^#^
*p* < 0.05, comparison between post-intervention period vs. baseline period.

**Table 3 microorganisms-13-00065-t003:** The number of multidrug-resistant microorganism infections.

	Baseline Period (Year 2018)	Intervention Period (Year 2019)	Post-Intervention Period (Year 2020)
Pathogen			
MRSA	2	1	0
VRE	11	0 *	2 ^#^
CRAB	14	4 *	7
CRPA	14	5 *	4 ^#^
CRE	7	3	5
Specimens			
sputum	23	11 *	12
urine	4	0 *	1
blood	14	2 *	4 ^#^
wound	1	0	1
Central line tip	2	0	0
Drain-tube tip	2	0	0
Pleural effusion	1	0	0
Ascites	1	0	0

* *p* < 0.05, comparison between intervention period vs. baseline period. ^#^
*p* < 0.05, comparison between post-intervention period vs. baseline period.

## Data Availability

The original contributions presented in this study are included in the article. Further inquiries can be directed to the corresponding author.

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
