# Peer review of "Implementation of 2% Chlorhexidine Bathing to Reduce Healthcare-Associated Infections Among Patients in the Intensive Care Unit"

_microorganisms, 2025, doi:10.3390/microorganisms13010065_

Round 1

Reviewer 1 Report

Comments and Suggestions for Authors

The authors investigated an important topic –  reducing HAI with antiseptic use, this paper investigate the implementation of  2% chlorhexidine bath. In an era of wide dissemination of pandrug and extremely drug resistant isolates the reduction of healthcare-associated pathogens is of big importance. This is a well written paper. The results were presented comprehensively and discussed well.

1.       Line 85 –in this section, please define the periods (which year)

2.       How does the author explain the slightly increased HAI in the post-interventional period since the use of a 2% chlorhexidine bath was not interrupted? Could this be related to  chlorhexidine resistance?

3.       In line 110/111 the authors defined HAI .  The same should be done for the group of patients with urinary tract infection and bloodstream infection. Please write the criteria for inclusion in these groups (for calculating the incidence) – for example, urine is from a patient with a catheter or without a catheter, whether the microbiological examination found microorganisms above 105 cfu/ml (real infection) or detect microorganisms in small amount. For BSI – patients have signs of sepsis or not. Do these cases include bacteremia without septic symptoms?

4.       Not only gram-negatives show increased resistance to chlorhexidine, but also MRSA (qacA/B and smr positive) - doi: 10.1128/AAC.00761-12.

Reviewer 2 Report

Comments and Suggestions for Authors

The topic is relevant to healthcare providers and policymakers. The title captures the content of the manuscript. The introduction orients the readers to the topic, discusses what is already known about the topic, and identifies the gap in the literature. The authors also state the aim of the study. The materials and methods clearly describe how the study was conducted to allow for its replication. The results are presented with the help of tables and figures. However, the discussion needs to be improved as the authors present new results of another meta-analysis.

MAJOR REVISIONS

Abstract

1. The authors should state the sample size and the sampling method used.

Introduction

2. The authors provided only one reference in lines 52-65. More references are required for this section.

3. In lines 71-72, the authors state, ‘However, the effects of catheter-associated urinary tract infections and Gram-negative bacteria are controversial.’ Can the authors provide more information on the controversies?

Materials and methods

4. The authors should provide references for lines 111-113.

Results

5. The results should be presented under subheadings such as participants’ characteristics, Hospital-acquired infections, etc.

Discussion

6. In lines 172-177, the authors present results of a meta-analysis when they added the current study. However, these are results of another study and, therefore, should not be included in this discussion. The meta-analysis is a stand-alone study which the readers were not informed about in the materials and methods section.

MINOR REVISIONS

7. In lines 227-228, the authors state, ‘However, adverse reactions to CHG involved immediate IgE-mediated type I hypersensitivity and ….’ These are reactions reported in previous studies so ‘involved’ should be replaced by ‘involve’

Round 2

Reviewer 2 Report

Comments and Suggestions for Authors

Thank you for the opportunity to review the manuscript again. The authors are mixing two studies. If they want to add a systematic review and meta-analysis, they must follow the guidelines of systematic review and meta-analysis in presenting their results. I suggest they remove the systematic review part and present it as a different paper. In this manuscript, let them concentrate on their primary study. Additionally, if they insist on presenting these 2 together, it should be stated in the introduction and the results of the systematic review should all be there, including characteristics of the studies, publication bias tests, heterogeneity tests, etc.

Author Response

Comments 1: Thank you for the opportunity to review the manuscript again. The authors are mixing two studies. If they want to add a systematic review and meta-analysis, they must follow the guidelines of systematic review and meta-analysis in presenting their results. I suggest they remove the systematic review part and present it as a different paper. In this manuscript, let them concentrate on their primary study. Additionally, if they insist on presenting these 2 together, it should be stated in the introduction and the results of the systematic review should all be there, including characteristics of the studies, publication bias tests, heterogeneity tests, etc.

Response 1: We totally agree with your comments. Although there is an example of incorporating a systematic review and meta-analysis into a clinical study (e.g. Tsai TY et al. JAMA Intern Med 2024; 184(1): 37-45). It seemed weird to combine a systematic review and meta-analysis result in an interventional study. We removed the systematic review part from the current manuscript. However, we sincerely hope that you can approve retaining the meta-analysis and forest plot in the discussion. The purpose of the forest plot is to reveal our efforts for a detailed literature review and just to compare the efficacy of the outcome in different studies, which can enrich the content in the discussion. If the forest plot is still your concern, we are happy to remove it in the subsequent revision.